# New Intracellular Peptide Derived from Hemoglobin Alpha Chain Induces Glucose Uptake and Reduces Blood Glycemia

**DOI:** 10.3390/pharmaceutics13122175

**Published:** 2021-12-16

**Authors:** Renée N. O. Silva, Ricardo P. Llanos, Rosangela A. S. Eichler, Thiago B. Oliveira, Fábio C. Gozzo, William T. Festuccia, Emer S. Ferro

**Affiliations:** 1Department of Pharmacology, Biomedical Science Institute, University of São Paulo, São Paulo 05508-000, SP, Brazil; oliveirarenee@gmail.com (R.N.O.S.); ricardopariona@hotmail.com (R.P.L.); reichlerusp@gmail.com (R.A.S.E.); 2Physiology and Biophysics, Biomedical Science Institute, University of São Paulo, São Paulo 05508-000, SP, Brazil; tbelchior12@gmail.com (T.B.O.); william.festuccia@usp.br (W.T.F.); 3Institute of Chemistry, State University of Campinas, Campinas 13083-862, SP, Brazil; fgozzo@gmail.com

**Keywords:** peptide drug discovery, bioactive peptides, glucose uptake, insulin signaling, diabetes

## Abstract

Intracellular peptides were shown to derive from proteasomal degradation of proteins from mammalian and yeast cells, being suggested to play distinctive roles both inside and outside these cells. Here, the role of intracellular peptides previously identified from skeletal muscle and adipose tissues of C57BL6/N wild type (WT) and neurolysin knockout mice were investigated. In differentiated C2C12 mouse skeletal muscle cells, some of these intracellular peptides like insulin activated the expression of several genes related to muscle contraction and gluconeogenesis. One of these peptides, LASVSTVLTSKYR (Ric4; 600 µg/kg), administrated either intraperitoneally or orally in WT mice, decreased glycemia. Neither insulin (10 nM) nor Ric4 (100 µM) induced glucose uptake in adipose tissue explants obtained from conditional knockout mice depleted of insulin receptor. Ric4 (100 µM) similarly to insulin (100 nM) induced Glut4 translocation to the plasma membrane of C2C12 differentiated cells, and increased GLUT4 mRNA levels in epididymal adipose tissue of WT mice. Ric4 (100 µM) increased both Erk and Akt phosphorylation in C2C12, as well as in epididymal adipose tissue from WT mice; Erk, but not Akt phosphorylation was activated by Ric4 in tibial skeletal muscle from WT mice. Ric4 is rapidly degraded in vitro by WT liver and kidney crude extracts, such a response that is largely reduced by structural modifications such as N-terminal acetylation, C-terminal amidation, and substitution of Leu8 for DLeu8 (Ac-LASVSTV[DLeu]TSKYR-NH2; Ric4-16). Ric4-16, among several Ric4 derivatives, efficiently induced glucose uptake in differentiated C2C12 cells. Among six Ric4-derivatives evaluated in vivo, Ac-LASVSTVLTSKYR-NH2 (Ric4-2; 600 µg/kg) and Ac-LASVSTV[DLeu]TSKYR (Ric4-15; 600 µg/kg) administrated orally efficiently reduced glycemia in a glucose tolerance test in WT mice. The potential clinical application of Ric4 and Ric4-derivatives deserves further attention.

## 1. Introduction

Evidence gathered over the years strongly supports the involvement of various proteases, peptidases, and peptides in the development of metabolic disorders such as insulin resistance, obesity, and metabolic syndrome [1,2,3,4,5]. Peptides, due to their remarkable potency, selectivity, and low toxicity, arise as good candidates for therapeutic applications [6,7]. Moreover, the complex structure of peptides, compared to that of small molecules, is no longer limiting their therapeutic use.

A peptide-capture assay using the catalytically inactive form of Thimet-oligopeptidase (EC 3.4.24.15; THOP1) allowed the seminal identification of a new class of functional intracellular peptides (InPeps) of therapeutic potential [8]. InPeps were suggested as products of proteasomal proteolysis [9,10,11,12,13], distinguishable from major histocompatibility class I (MHC-I) antigens, neuropeptides, and/or cryptides [9,10,11]. Different treatments and/or diseases modify the relative concentration of specific InPeps present in cells or tissues, suggesting pathophysiological functions [10,14,15]. Pharmacological and commercial applications of InPeps have been previously suggested [16]. In challenge conditions, cells accumulate or lose specific InPeps that are biologically functional [17,18,19]. Possible biological functions for InPeps have been suggested to take place either intracellularly, through modulation of protein interactions [20,21] and microRNA stability [17], or extracellularly through binding to plasma membrane receptors [20,22,23,24,25,26]. Mice with whole-body deletion of THOP1 were resistant to diet-induced obesity and insulin resistance after being fed a high fat diet for 24 weeks [17]. Moreover, the relative levels of at least four InPeps were increased in C57BL6/N null mice for the endopeptidase neurolysin (EC 3.4.24.16; Nln^−^^/−^) [4], which display improved glucose tolerance and insulin sensitivity. Altogether, these findings suggest the exciting possibility that InPeps may protect mice from diet induced obesity and insulin resistance [4]. Two of these relatively increased InPeps were identified from the gastrocnemius muscle to derive from troponin I (SADAMLKALLGSKHK, Ric1; DMEVKVQKSSKELEDMNQKL, Ric2). Another two InPeps that were relatively increased in Nln^−/−^ were from the epididymal adipose tissue, one being derived from acyl-CoA binding protein (VEKVDELKKKYGI, Ric3) and the other one derived from hemoglobin alpha subunit (LASVSTVLTSKYR, Ric4) [4].

Antidiabetic agents such as thiazolidinediones and biguanide metformin enhance insulin sensitivity, and effectively reduces hyperglycemia [27]. However, these drugs show several undesired side-effects after long-term administration [28], emphasizing the need for the development of new compounds to treat these conditions. In this sense, peptides due to their remarkable potency, selectivity, and low toxicity, arise as good candidates [7]. Indeed, previous studies have indicated that peptides are key regulators of various cellular and intercellular physiological responses and therefore possess an enormous potential for the treatment of various diseases including obesity and insulin resistance [7].

Here, we investigated the effects on glucose uptake and homeostasis of the InPeps Ric1, Ric2, Ric3, and Ric4, which were increased in glucose tolerant and insulin sensitive Nln^−/−^ mice [4]. Our hypothesis was that Ric1, Ric2, Ric3, and/or Ric4 could function synergically or similarly to insulin inducing glucose uptake and reducing glycemia.

## 2. Experimental Procedures

### 2.1. Animals

In vivo experiments were conducted with male C57BL/6N (WT) mice 12–16 weeks-old (University of São Paulo Medical School Animal Facility, SP, Brazil). Mice were maintained in individual ventilated cages (Amresco, SP, Brazil), under standardized conditions with an artificial 12 h dark–light cycle, with free access to standard chow (Nuvital Nutrientes S.A., Colombo, PR, Brazil) and drinking water ad libitum. The number of animals used was the minimum necessary to obtain statistically significant results and they were maintained and used in accordance with the guidelines of the National Council for Control of Animal Experiments (CONCEA), following international norms of animal care and maintenance. Thus, we hereby state that all experimental protocols were previously approved by University of São Paulo Ethic committee councils from Biomedical Science Institute (approval number for mice experimentation ICB/USP 22/2017).

### 2.2. Glucose Tolerance Tests (GTT)

WT mice were fasted for 12 h and blood samples were taken before and at 15, 30, 60, 90, and 120 min after the intraperitoneal (ip) injection of 2 g/kg glucose [29,30]. Glycemia was measured using a glucometer (Accu-Check Performa, ROCHE, São Paulo, Brazil). Peptides were evaluated to affect glucose uptake either after ip or oral (by gavage) administration. After 20 min glucose was administrated (blood glucose levels reached 400 mg/dL), animals received ip or oral administration of either saline, insulin (0.75 IU/kg), or peptide (Ric2 or Ric4; 600 µg/kg). In addition, a group of animals received oral administration of either water or Ric2 or Ric4 (600 µg/kg). Glycemia was measured as indicated following vehicle, insulin, or peptide administration [17].

### 2.3. Cell Culture

C2C12 cells were used as a well-described model of skeletal muscle to evaluate the effect of peptides in gene expression, cell signaling, and glucose uptake [29,31,32,33,34]. These cells were kindly provided by Prof. Patricia Brum, School of Physical Education and Prof. Anselmo Moriscot, Biomedical Science Institute, University of São Paulo, São Paulo, Brazil. C2C12 cells were cultured and maintained in high glucose Dulbecco’s modified Eagle’s medium (DMEM; Thermo Fisher Scientific, São Paulo, SP, Brazil) supplemented with 10% fetal bovine serum (FBS), 100 U/mL penicillin, and 100 mg/mL streptomycin in a humidified atmosphere of 5% CO_2_ at 37 °C. The cells were used in low pass (between passages 2 and 4) for all experiments to maintain the potential for differentiation of cultures. Cells grown to approximately 70% confluence in T75 culture flasks were treated with trypsin and seeded in various P100 culture dishes. Cells were then cultured in the presence of 10% FBS until they reach about 90% confluence, at which point the medium was replaced with DMEM containing 2% horse serum for the induction of differentiation in myotubes. During this time, the fused myoblasts formed elongated, multi-nucleated myotubes, and within 7 days more than 95% of the cells already fused into myotubes. Cultures of C2C12 were treated with phosphate buffer saline (137 mM NaCl, 2.7 mM KCl, 10 mM Na_2_HPO_4_, and 1.8 mM KH_2_PO_4_, pH 7.4; PBS; control), insulin or InPeps, for the indicated period of time. From these treatments the cells were processed for molecular analyses.

### 2.4. Cell Viability

Cell viability tests were performed using 3-(4,5-dimethizzol-zyl)-2-5-diphenyl tetrazolium bromide (MTT). The purpose of these tests was to observe whether the peptides evaluated have an effect on the cellular viability. MTT is a yellow, water-soluble reagent. It can penetrate the cells through the membrane and, in contact with the superoxide produced only by the mitochondrial activity of living cells, is oxidized to MTT-formezan (a salt of purple color and insoluble in water). Thus, only living cells acquire purplish staining [35]. To perform these cell viability tests, C2C12 cells were incubated in the presence of different peptide concentrations for up to 6 h. After incubation, the medium was replaced with serum-free and antibiotic-free DMEM plus 10% MTT reagent (5 mg/mL) and incubated at 37 °C for 1 h. Thereafter, the medium was replaced with a solution of isopropyl alcohol/0.04 N HCl and stirred vigorously. The supernatant was removed and the absorbance read at wavelength of 570 nm [35].

### 2.5. Western Blotting

Western blotting experiments were performed to evaluate the expression levels of specific proteins. Briefly, cells or tissues as indicated were solubilized by directly adding RIPA buffer (50 mM Tris HCl, 150 mM NaCl, 1.0% *v*/*v*, NP-40, 0.5% *w*/*v*, sodium deoxycholate, 1.0 mM ethylenediamine tetraacetic acid, EDTA, 0.1% *w*/*v*, sodium dodecyl sulphate, SDS, and 0.01% *w*/*v* sodium azide, pH of 7.4) containing protease (P8340; Sigma Aldrich, Sao Paulo, Brazil) and phosphatase (P5726, Sigma Aldrich, Sao Paulo, Brazil) inhibitors cocktails, followed by sonication with a micro-tip for 5 s. After 30 min of incubation on ice, samples were centrifuged for 30 min at 12,000× *g*. Proteins from supernatants (50 µg) were separated by 10% SDS-PAGE and transferred to nitrocellulose membranes for immunoblotting with mouse monoclonal anti-phospho extracellular signal-regulated kinase 1/2 (anti-pErk; Erk 1/2 phosphorylation at Thr202-Tyr204; 1:1000; Cell Signaling, Boston, MA, USA), and rabbit polyclonal anti-total extracellular signal-regulated kinase 1/2 (Erk; anti-total Erk 1/2, 1:1000; Cell Signaling, Boston, MA, USA) antibodies. Blots were also treated with rabbit anti-phospho Akt (S473) (anti-pAkt, 1:1000) and mouse polyclonal anti-total Akt (anti-Akt, 1:1000) (Cell Signaling, Boston, MA, USA) and anti-myogenin (1:1000, Abcam, UK). Protein normalization indicated was performed using rabbit polyclonal anti-glyceraldehyde 3-phosphate dehydrogenase (GAPDH; 1:2000; Proteimax Biotechnology, São Paulo, Brazil) antibodies or anti-β-actin mouse antiserum (1:2000; Sigma Aldrich, Sao Paulo, Brazil). Blots were incubated with horseradish peroxidase-conjugated secondary antibodies for 3 h at room temperature. Western blot bands were visualized with Super-Signal West Pico Chemiluminescent substrate (Thermo Scientific) using the ChemiDoc™ MP Imaging System (BioRad, Hercules, CA, USA), and quantified using ImageJ 1.49 software. All results are expressed as the means ± standard error of the mean (SEM). The statistical comparisons were performed using Student’s *t*-test or analysis of variance (ANOVA), followed by ad-hoc Tukey’s test (*n* = 7). Data were statistically analyzed with GraphPad Prism software (GraphPad Software Inc, San Diego, CA, USA).

### 2.6. Peptide Synthesis

Peptides were synthesized using Fmoc (N-(9-fluorenyl) methoxycarbonyl) chemistry and were further purified by high-performance liquid chromatography (HPLC) to ≥95% purity (Table 1; Proteimax Biotechnology LTDA, São Paulo, SP, Brazil), similarly to previously described [20]. For all experiments, peptides stock solutions (10 mg/mL) were prepared in autoclaved Milli-Q water and kept at −80 °C. Additional dilutions were prepared immediately before the experiments in the appropriate vehicle (i.e., DMEM, phosphate buffered saline, or autoclaved Milli-Q water).

### 2.7. Intracellular Peptide Stability

In order to determine if the peptides investigated herein were substrates of tissue peptidases, 50 μM of each peptide was individually incubated for 20 min in the presence of the crude tissue extract (liver, 30 µg; kidney, 3 µg) in a final volume of 250 μL of 0.025 M Tris-HCl, containing 0.125 M NaCl (TBS). Peptide hydrolyses were analyzed by reverse phase liquid chromatography (HPLC) using a C18 µBond-pak column (4.6 × 250 mm; Millipore Corp.) with a linear gradient of 5–65% acetonitrile in 0.1% TFA for 20 min at a flow rate of 1 mL/min, and absorbance monitored at a wavelength of 214 nm, as previously described [8,36,37]. Cleaved peptide bonds were identified by mass spectrometry sequencing after isolating the cleavage fragments manually after HPLC, as previously described [8,38] (data not shown). Control enzyme assays were conducted using bradykinin (RPPGFSPFR; 50 µM) as a standard substrate [37].

### 2.8. Real-Time PCR

Real-time PCR (qPCR) experiments were performed to evaluate the mRNA expression levels, at basal and under stimulus, for peroxisome proliferator activator receptor gamma (PPARγ), peroxisome proliferator activator receptor alpha (PPARa), cAMP responsive element binding protein 1 (Creb1), aldolase A, fructose-bisphosphate (ALDOA), cytochrome c oxidase subunit IV isoform 1 (Cox4i1), phosphoglycerate mutase 2 (PGAM2), troponin I, skeletal, fast 2 (TNNI2), troponin I, skeletal, fast 3 (TNNT3), small muscle protein X-linked (SMPX), myosin, light chain 1 (MYH1), insulin-growth factor 1 (IGF1), Glucose transporter 4 (GLUT4), and lipoprotein lipase (LPL). Primers sequences are described in Appendix A. Cells were homogenized and total RNA extracted using Trizol according to the manufacturer’s instructions (TRIzol, Life Technologies, Rockville, MD, USA). After extraction of RNA, all samples were purified and treated with DNAse using RNeasy Mini Kit (Qiagen, Foster City, CA, USA). The integrity of the RNA was verified by 1% agarose gel electrophoresis stained with ethidium bromide and visualized in ultraviolet light. The cDNAs were synthesized from 2 µg of total RNA with Moloney Murine Leukemia Virus Reverse Transcriptase (Invitrogen, Carlsbad, CA, USA) using random hexamer nucleotides. The standard curves for all the primers used herein were previously constructed to determine the efficiency of the amplification of the target and reference genes. Quantitative PCRs were performed using the Prism 7900 sequence detection system (Applied Biosystems, Foster City, CA, USA) with 100 nM primer and 20 ng cDNA. The expression of mRNA targets was normalized by expression of gene glyceraldehyde 3-phosphate dehydrogenase (GAPDH) or hipoxantina-guanina fosforibosiltransferase (HPRT) and expressed as relative values using the 2 DDCt method [39]. The expression levels of target genes were normalized with housekeeping genes GAPDH or HPRT, and alterations were expressed relative to control unstimulated cells.

### 2.9. Glucose Uptake in Adipose Explants

Glucose uptake was evaluated in adipose tissue explants from mice bearing or not deletion of insulin receptor in adipocytes essentially as previously described [40]. Briefly, explants of epididymal adipose tissue (20–25 mg) were incubated in 1 mL of Krebs–Ringer bicarbonate buffer (in mmol/L): 118 NaCl, 4.8 KCl, 1.25 CaCl_2_, 1.2 KH_2_PO_4_, 1.2 MgSO_4_, 25 NaHCO_3_, and 5.5 glucose and 1 μCi/mL of ^3^H-deoxyglucose (New England Radiochemicals, Boston, MA, USA) supplemented with 2% fatty acid-free BSA (Sigma, Oakville, ON, Canada), 7.4. Vials were incubated at 37 °C for 1 h in the presence or absence of insulin (100 nM) or Ric4 peptide (100 µM). After 30 min, explants were extensively washed and processed to evaluate the uptake of ^3^H deoxyglucose as previously described [40].

### 2.10. Glucose Uptake Assays

C2C12 myotubes were incubated for 3 h with serum-free, 5 mM glucose-DMEM, then incubated with Hepes buffer for 30 min and then incubated for further 30 min in glucose-uptake buffer containing vehicle, Insulin (100 nM), Ric4 or Ric4 derived peptides (100 μM) in the presence of ^3^H-glucose (1uCi/mL; New England Radiochemicals, MA, USA)or 2-[N-(7-Nitrobenz-2-oxa-1,3-diazol-4-yl)amino]-2-deoxy-d-glucose (2-NBDG/80 μM; (Invitrogen, Carlsbad, CA, USA). Plates were incubated at 37 °C with 5% CO_2_ for a period of 30 min. After that, cells were lysed with 50 μL of 0.1 N NaOH and fluorescence or beta radiation of aliquots from the lysate was measured in spectrophotometer or beta counter, respectively and normalized by mg of total protein [41].

### 2.11. GLUT4 Translocation Assays

C2C12 differentiated myotubes were starved with free-FBS low glucose DMEM for 3 h, and then treated with saline or 5 µM of BMS-536924 insulin receptor inhibitor (Sigma-Aldrich, Sao Paulo, Brazil) for 30 min. Next, C2C12 were incubated in the presence of either vehicle, insulin (100 nM) or Ric4 (100 µM) for 30 min. As described by Tortorella and Pilch, 2002, cells were washed three times with PBS, and homogenized with 40 strokes of a glass tissue grinder in buffer containing sucrose 255 mM, disodium 4 mM, EDTA 20 mM, 4-(2-hydroxyethyl)-1-piperazineethanesulfonic acid (HEPES) 10 mM, pH 7.4, leupeptin 10 µM, pepstatin 1µM, aprotinin 1 µM, phenylmethylsulfonyl fluoride (PMSF) 1 mM, and benzamidine 5 mM (HES buffer). The homogenate was centrifuged at 19,000× *g* for 20 min. The pellet was saved (P1). The supernatant was recovered and centrifuged at 40,000× *g* for 20 min. The pellet (MF1) was resuspended in HES buffer. The supernatant was further centrifuged at 180,000× *g* for 1.5 h and the pellet (MF2) was resuspended in HES buffer. Both pellets MF1 and MF2, referred to as microsomal fraction (MF) correspond to high- and low-density microsomes. The pellet from the first centrifugation (P1) was resuspended in HES buffer, layered onto a 1.12 M sucrose cushion in 20 mM HEPES and 1 mM disodium EDTA, and centrifuged at 100,000× *g* for 1 h. The pellet (nuclear/endoplasmic reticulum fraction -N/ER) was resuspended in a buffer containing 20 mM Tris (pH 7.4), 50 mM NaCl, 2% Nonidet P-40, 0.5% deoxycholate, 0.2% SDS, and the protease inhibitor cocktail (#P8340; Sigma Aldrich, SP, Brazil). The interphase of the sucrose cushion was collected and pelleted at 40,000× *g* for 20 min. This PM-containing pellet (PM) was resuspended in PBS plus protease inhibitors. All centrifugations were performed at 4 °C with a Beckman Coulter Ultracentrifuge [42]. Total protein of all samples was determined using Pierce™ BCA Protein Assay Kit (ThermoFisher, Sao Paulo, Brazil). Proteins were submitted to Western blot analysis as described above, using as primary antibody anti-Glut4 (1:1000, Cell Signaling, Boston, MA, USA).

### 2.12. Statistical Analyses

All results were expressed as mean ± SEM. Statistical analyses were conducted by unpaired t test for independent samples, or analysis of variance (ANOVA) followed by ad-hoc Tukey’s test and/or Bonferroni test for samples comparing 3 or more groups. Values of p considered significant: * *p* < 0.05; ** *p* < 0.01 or *** *p* < 0.001.

## 3. Results

Myotubes from differentiated C2C12 cells were used as a cell model to evaluate the effects of selected InPeps (Ric1-Ric4) in gene expression, cell signaling, and glucose uptake (Figure 1).

MTT tests indicate that peptides Ric1-Ric4 (100 µM) were not triggering toxicity in differentiated C2C12 myotubes (data not shown). Therefore, the possible effects of Ric1-Ric4 (100 µM) on gene expression were evaluated. For all these experiments, insulin was used as “gold standard”/positive control, and GAPDH expression was used as normalization control (Table 2, Control “1”/100%). Relative to GAPDH expression, insulin increased the expression of all genes evaluated (Table 1). Ric1, Ric 2, and Ric4, but not Ric3, increased the relative expression of only specific genes related to glucose metabolism via glycolysis (i.e., aldolase and phosphoglycerate mutase) and skeletal muscle contraction (i.e., troponin I, small muscle protein X-linked, myosin; Table 2). These results suggested specificity and therapeutic potential of Ric1, Ric2, and Ric4 peptides to regulate the expression levels of specific genes.

Next, the effects of Ric1, Ric2, Ric3, and Ric4 on cell signaling were investigated by Western blotting through phosphorylation of Erk and/or Akt, in differentiated C2C12 cells (Figure 2). Insulin increased the phosphorylation of both Erk (pErk) and Akt (pAkt; Figure 2A,B). Ric2 and Ric4 activated pErk (Figure 2A), while Ric2, Ric3 and Ric4 activated pAkt (Figure 2B). These data corroborate previous suggestions that both Erk and Akt signaling pathways can be activated by insulin [43]. Insulin activation of Akt phosphorylation and signaling pathways have been most frequently associated with Glut4-induced glucose uptake [44]. Therefore, Ric2, Ric3, and Ric4 could have the potential to induce glucose uptake.

Next, glucose tolerance test (GTT) was used to investigate in vivo pharmacological effects of Ric2 and Ric4 on glucose homeostasis [4]; these two peptides were chosen because they altered both gene expression and Akt phosphorylation. Animals received an ip administration of either saline, insulin (0.75 IU/kg), Ric2 (600 µg/kg), or Ric4 (600 µg/kg), 20 min after glucose administration. Both Ric2 and Ric4 ip administrated, rapidly induced a decrease in blood glucose levels in WT mice (Figure 3A,B). Ric4 also decreased blood glucose levels in the GTT following oral administration by gavage (Figure 3C,D). Therefore, because Ric4 peptide showed activity both orally and ip, it was chosen for additional pharmacological characterizations considering its potential therapeutic application in the future.

Similar to insulin, Ric4 was observed to stimulate glucose uptake both in differentiated C2C12 (Figure 4A; left panel, using 3H-glucose; right panel, using 2-NBDG H-glucose) and adipose tissue explants obtained from C57BL6N WT mice (Figure 4B). Conversely, neither insulin nor Ric4 were capable of inducing glucose uptake from adipose tissue explants obtained from conditional knockout mice depleted from insulin receptor (Figure 4B). Moreover, Ric4 distinctively from insulin increased the translocation of Glut4 to rough endoplasmic reticulum (RER) membranes (Figure 4C), decreasing Glut4 presence in microsomal membranes (Figure 4D). Similar to insulin, Ric4 induced translocation of Glut4 to plasma membrane (Figure 4E), suggesting a mechanism of action in stimulating glucose uptake.

Ric4 ip administration in WT mice increased Erk and Akt phosphorylation on epididymal adipose tissue (Figure 5A,B). On tibial skeletal muscle, Ric4 increased Erk but not Akt phosphorylation (Figure 5C,D).

Ric4 (600 µg/kg) ip administration in WT mice stimulated the expression levels of genes related to energy metabolism in the epididymal adipose tissue (Figure 6A–E), whereas similar results were not observed in the tibial skeletal muscle tissue (Figure 6F–H). Together, in vivo results corroborate the pharmacological activity and therapeutic potential of Ric4.

Rapid degradation of Ric4 by WT mice liver or kidney crude extracts suggested that in vivo an initial enzymatic cleavage occurred on the Leu8-Thr9 peptide bond (L1A2S3V4S5T6V7L8T9S10K11Y12R13; data not shown). Structural modifications including N-terminal acetylation, C-terminal amidation and substitution of Leu8 for DLeu8 (Ac-LASVSTV[DLeu]TSKYR-NH2; Ric4-16) largely reduced the relative degradation ratio of Ric4 (Table 2; Appendix A); bradykinin was used a standard peptide. Alone, neither N-terminal acetylation nor C-terminal amidation protected Ric4 from degradation, either by liver or kidney tissue homogenates; conversely, such modifications largely increased Ric4 degradation by kidney extracts (Table 2). The relative degradation ratio of Ric4 was also evaluated by recombinant THOP1 and neurolysin (Nln) [37]. Ric4 was relatively a poor substrate for both THOP1 and Nln, compared to bradykinin (Table 2). On the other hand, Ric4-16 was a better substrate of both Nln and THOP1 compared to Ric4 or bradykinin (Table 2).

To gain further insight on Ric4 structure-activity related to glucose uptake, several Ric4 derived peptides were designed (Table 3). The rationale to design additional peptides was to identify a minimal Ric4-derived sequence retaining the ability to induce glucose uptake. Thus, Ric4 original sequence was successively shortened from amino and/or carboxyl terminal.

Using differentiated C2C12 cells, Ric4-derived peptides were evaluated regarding the ability to induce glucose uptake (Figure 7). Indeed, in C2C12 cells several Ric4-derived peptides efficiently induced glucose uptake in C2C12 cells (Figure 7).

Next, GTT were conducted to evaluate the in vivo effect of selected Ric4-derived peptides (Ric4-1, Ric4-2, Ric4-6, Ric4-14, Ric4-15, or Ric4-16) in glucose uptake, following oral administration to C57BL6N WT mice (Figure 8). Peptides selected for these assays included both Ric4-1, unable to induce glucose uptake, and derivatives highly active on inducing glucose uptake in C2C12 cells. Ric4-2 (Ac-LASVSTVLTSKYRNH2) and Ric4-15 (Ac-LASVSTV[DLeu]TSKYR) (600 μg/kg) were the two Ric4-derived peptides to show ability to reduce blood glucose levels following oral administration (Figure 8B,E). These results suggested that yet nonidentified structural features of Ric4 were relevant for in vivo activity on glucose homeostasis. Possibly, more than one single modification (i.e., preventing Leu8-Thr9 endopeptidases degradation, or amidation of carboxyl terminus to prevent carboxypeptidases degradation) affected Ric4 pharmacological activity/pharmacokinetics in vivo.

## 4. Discussion

The main results presented herein suggest that Ric4, a natural InPep derived from hemoglobin alpha chain, has several pharmacological activities similar to insulin. Ric4 activated both Erk and Akt phosphorylation in differentiated C2C12 cells and in adipose tissue from WT mice. Ric4 was not capable of inducing glucose uptake from adipose tissue explants obtained from conditional knockout mice depleted from insulin receptor. These data suggest that Ric4 needs the insulin receptor to promote glucose uptake. Ric4 was rapidly metabolized in vitro by enzymes present in liver and kidney tissue extracts, which could be largely reduced after modifying Ric4 original structure. Some of these Ric4 derivatives retain the ability to induce glucose uptake and reduce glycemia following oral administration to WT mice. The results presented herein successfully suggest that InPeps such as Ric4 have biological and pharmacological significance. However, possible clinical applications of Ric4 still deserves further investigation.

Insulin is a potent peptide hormone, essential for the maintenance of glucose homeostasis and for cell growth and development [45,46]. This hormone is secreted by β-cells from the islets of Langerhans of the pancreas in response to increased circulating levels of glucose and amino acids after meals [47,48,49]. Insulin regulates glucose metabolism by reducing hepatic glucose production (through inhibition of glycogenolysis and gluconeogenesis) and increasing glucose uptake, primarily in skeletal muscle and adipose tissue. These insulin actions are mediated via the PI3K/Akt signaling pathway [49,50]. Activation of the PI3K/Akt pathway begins when insulin receptor substrates (IRS) IRS1 and IRS2 are phosphorylated by the insulin receptor. These specific tyrosine phosphorylated sites serve as anchoring for the p85 regulatory subunit of the PI3K enzyme, through its SH2 binding domain [51]. Glucose uptake in skeletal muscle and adipose tissue is performed through the glucose transport protein 4 (Glut4) [52]. Glut4 belongs to a family of transport facilitators comprising 12 different types, and Glut4 is the only carrier of this family predominantly located in intracellular compartments [53]. Activation of PI3K-Akt by insulin induces the translocation of Glut4 to the plasma membrane, consequently increasing the uptake of glucose [53]. Following oral administration, Ric4 enhanced Glut4 mRNA levels in the adipose tissue of WT mice. Similar to insulin, Ric4 promoted translocation of Glut4 to plasma membrane on differentiated C2C12 skeletal muscle cells, which possesses the basic machinery required for translocation of GLUT4 in response to insulin stimulation [54]. Ric4 also increased Glut4 immunoreactivity on RER membranes and reduced Glut4 immunoreactivity on microsomal membranes, which were not observed for insulin herein. Additional distinctive activities of Ric4 compared to insulin includes a restricted action preferentially stimulating the expression of genes related to skeletal muscle contraction over glycolysis. Therefore, binding to insulin receptor could provide a mechanism of action for Ric4, such as on gene activation, activation of Erk and Akt signaling pathways, translocation of Glut4 to plasma membrane, increased glucose uptake and reduction of glycemia. Further data supports that the insulin receptor could be the biological/pharmacological target of Ric4, which was not able to promote glucose uptake in adipose tissue explants obtained from mice lacking the insulin receptor. However, Ric4 have some pharmacological distinctions from insulin such as on gene expression and Glut4 translocation to rough endoplasmic reticulum and microsomes. Previous reports proposed the existence of bias agonists for various tyrosine kinase receptors [55,56,57], including the insulin receptor family [58,59,60]. Flavonoids and triterpene alisol A-24-acetate (AA-24-a) also seem to bind to insulin receptor to mimic insulin action on glucose uptake and blood glycemia [34,61,62]. The soybean peptide aglycin regulates glucose homeostasis by enhancing insulin signal at gene levels of insulin receptor and IRS1, and the number of Glut4 at the cell surface [28]. Therefore, further investigations should be conducted to characterize the biological/pharmacological target and the mechanism of action of Ric4.

Ric4 corresponds to the last 13 C-terminal amino acids of mice hemoglobin alpha chain, and was relatively increased in the adipose tissue of Nln^−/−^ compared to WT mice [4]. Ric4, similar to previous hemoglobin-derived peptides, could be produced within erythrocytes from proteasome digestion of hemoglobin [63,64]. Secreted intracellular peptides correspond to approximately 10% of the total number of peptides identified in mouse brain tissue cultures [22,65]. These secreted cytosolic bioactive peptides have been proposed to be “non-classical neuropeptides”, as they can be synthesized in brain and secreted as bioactive entities, in a regulatory manner [22,65]. Ric4 could be secreted by erythrocytes to regulate glucose uptake in vivo. Increased levels of Ric4 were observed in adipose tissue of Nln^−/−^, which were animals shown to have increased glucose uptake and insulin sensitivity compared to WT mice [4]. Therefore, the present data suggest that Ric4 could contribute to the enhanced glucose uptake and insulin sensitivity observed in Nln^−/−^. Peptide Ric4-6 shown here to stimulate glucose uptake in C2C12 cells, was previously found in mice blood and could be stimulating glucose uptake systemically [66]. Therefore, it is tempting to suggest that Ric4 have biological significance, and together with some of its derivatives it could be participating of homeostatic control of blood glucose. However, the exact physiological mechanism still deserves further investigation.

Ric4 has unexpected in vivo ip and oral pharmacological activity, decreasing blood glucose levels in GTT, despite being rapidly degraded in vitro. A number of previous reports described ip and/or orally active peptides [16], including hemopressin [26], Pep19 [25], peptides C111/C112 from liver of *Katsuwonus pelamis*, peptides IPP and VPP and tryptic peptides from casein [67,68,69]. The mechanism(s) allowing specific peptides to escape proteolytic degradation, being absorbed by digestive system, remains yet unknown. Ric4 was rapid degraded by WT mice liver or kidney crude extracts, whereas structural modifications including N-terminal acetylation, C-terminal amidation, and substitution of Leu8 for DLeu8 (Ac-LASVSTV[DLeu]TSKYR-NH2; Ric4-16) largely reduced the relative degradation ratio of Ric4. Similar proteolytic stability was not observed for Ric4-2 lacking [DLeu] modification. Ric4, Ric4-2, and Ric4-15 were pharmacologically active following oral administration to WT mice. Ric4-16 shows no oral bioavailability, despite being proteolytically more stable than Ric4-2. These data suggest that proteolytical stabilization alone was not sufficient to allow greater bioavailability to Ric4. Additional experiments are underway to improve the proteolytic stability of Ric4, as well as to associate it with a new formulation that could improve its bioavailability.

## 5. Conclusions

In conclusion, Ric4 was successfully characterized herein as a new pharmacologically active peptide. The possibility that Ric4 has physiological significance, regulating glucose homeostasis, cell signaling, and gene expression was also suggested herein. Additional studies should be conducted to allow a better understanding of Ric4 biological and pharmacological mechanisms of actions, focusing on its possible clinical application. Indeed, in the past 5–6 years the U.S. Food and Drug Administration (FDA) have authorized the clinical prescription of more than 15 new peptides or peptide-containing molecules [6,70]. These data suggest that large scale production of peptides in addition to the complex structure of peptide-based drugs, compared to that of small molecules, have slowly been becoming no impeditive to the pharmaceutical industry. Therefore, exciting market perspectives exist for the development of Ric4-based drugs.

## Figures and Tables

**Figure 1 pharmaceutics-13-02175-f001:**
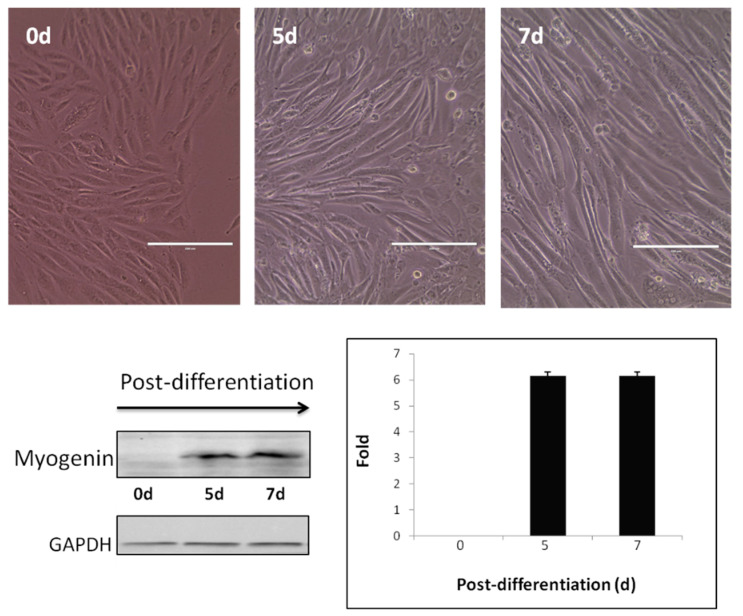
Myotubes from differentiated C2C12 cells. Upper panels show C2C12 mouse skeletal muscle myocyte cells (day 0; 0d) grown in DMEN containing 10% fetal bovine serum (FBS). After reaching a confluence of 80–90%, C2C12 cells were switched to DMEN containing 2% FBS and incubated for a period from 5 (5d) to 7 (7d) days, until myotubules could be observed. Morphological changes observed on C2C12 cells were confirmed at the molecular levels by Western blotting assays using an anti-myogenin antibody (left lower panel); the presence of myogenin only occurs in differentiated cells. Constitutive expression of GAPDH protein was used to normalize and quantify the relative expression of myogenin along the period of differentiation (right lower panel). Scale bars, 10 µM.

**Figure 2 pharmaceutics-13-02175-f002:**
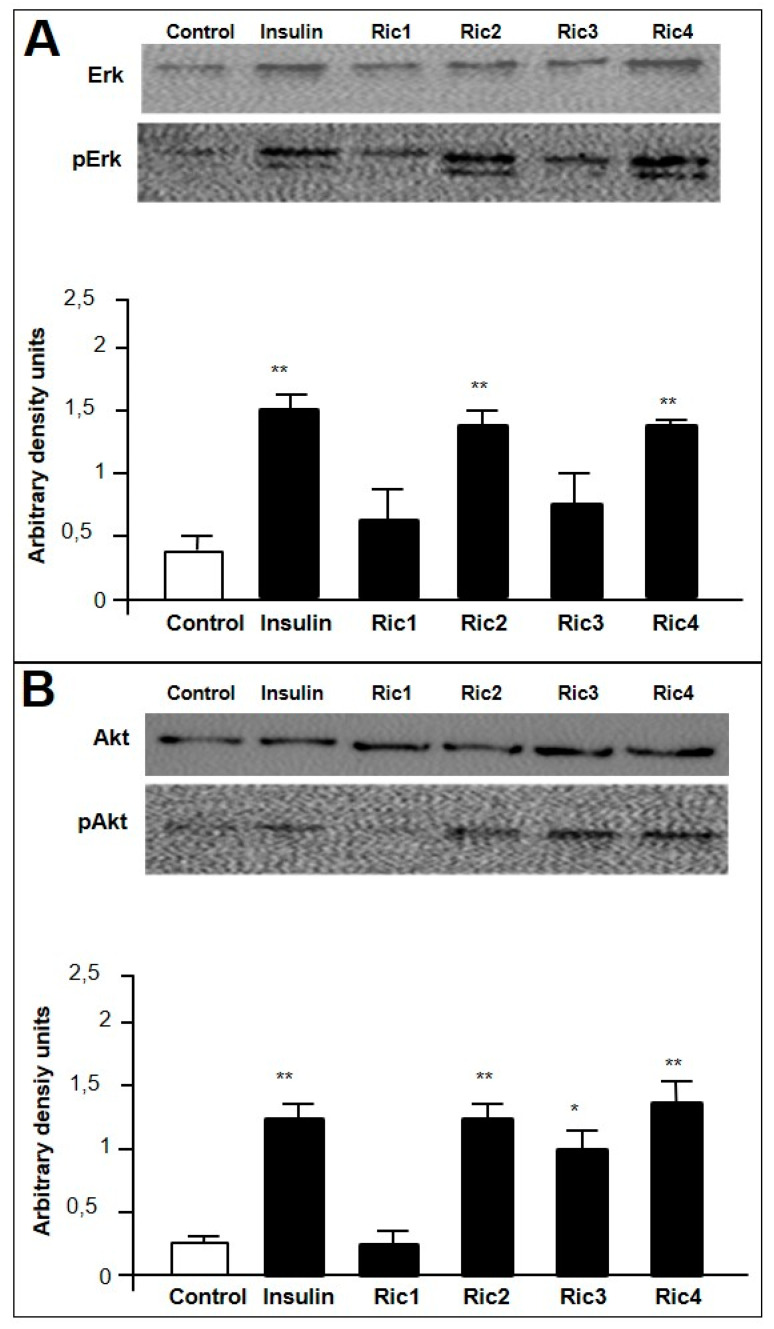
C2C12 cell signaling assays by Western blotting. C2C12 cells previously differentiated into myotubes were treated in the presence or absence of insulin (100 nM), or the indicated intracellular peptide (Ric1, Ric2, Ric3, or Ric4; 100 µM). Induced phosphorylation of Erk (**A**) or Akt (**B**), respectively, pAkt or pErk, were analyzed by Western blotting as described in Section 2.5. Imaging and band intensity measurements were performed using the ImageJ 1.49 software, and quantifications were performed evaluating the relative levels of pErk or pAkt over total Erk or Akt, respectively. The effect of the indicated peptide on the relative phosphorylation levels were expressed using arbitrary density units (**A**,**B**, lower panels). Data are representative of three independent experiments that produced similar results. The statistical comparisons were performed using Student’s *t*-test or analysis of variance (ANOVA), followed by ad-hoc Tukey’s test using GraphPad Prism software * *p* < 0.05; ** *p* < 0.001.

**Figure 3 pharmaceutics-13-02175-f003:**
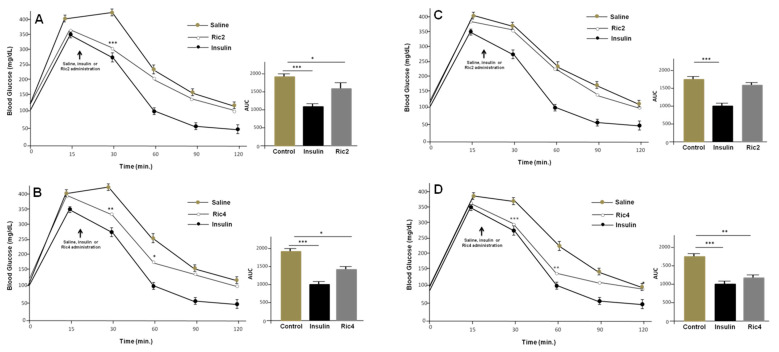
Glucose tolerance tests (GTT) in WT mice administrated ip or orally with peptides Ric2 or Ric4. The animals were fasted for 12 h before the treatments. Glucose 2 g glucose/kg was injected at time zero and at 20 min with insulin (0.75 IU/kg) or Ric2 or Ric4 (100 µM). (**A**,**B**), ip administration of Ric 2 or Ric4; (**C**,**D**), oral administration of Ric 2 or Ric4. Control animals were administrated with saline (ip) or water (oral). The statistical comparisons were performed using Student’s *t*-test or analysis of variance (ANOVA), followed by ad-hoc Tukey’s test using GraphPad Prism software * *p* < 0.05; ** *p* < 0.001; *** *p* < 0.0001.

**Figure 4 pharmaceutics-13-02175-f004:**
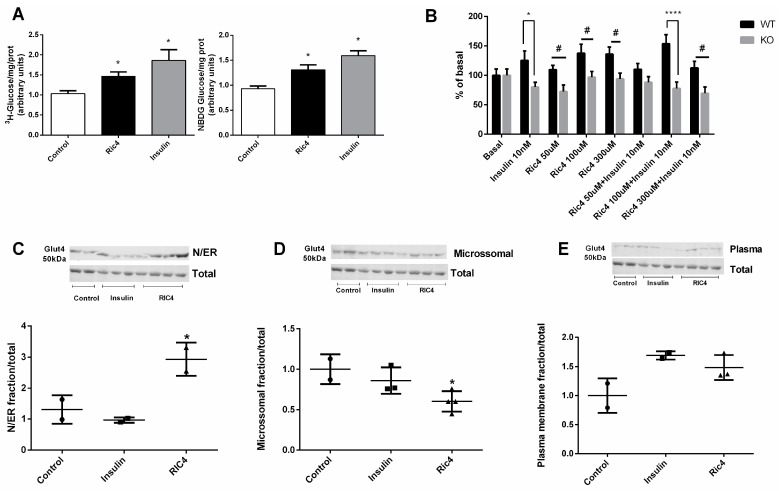
Glucose uptake and GLUT4 translocation induced by Ric4 in C2C12 cells. C2C12 cells were previously incubated in serum-free and glucose-free DMEM medium, then incubated with HEPES buffer for 30 min and then incubated for a further 30 min in glucose-uptake buffer containing vehicle, Ric4 (100 µM), or insulin (100 nM) in the presence of ^3^H-glucose (1 µCi/mL) or 2-NBDG (80 µM). After incubations cells were lysed with 50 μL of 0.1 N NaOH and fluorescence or radiation of aliquots from the lysate were measured (**A**). Epididymal adipose tissue explants (20–25 mg) were incubated in Krebs–Ringer bicarbonate buffer containing glucose 5.5 mM and 1 μCi/mL of ^3^H-deoxyglucose supplemented with 2% fatty acid-free for 30 min at 37 °C in the presence or absence of insulin (100 nM) or Ric4 (100 µM). The explants were processed to evaluate the uptake of ^3^H-deoxyglucose (**B**). Myotubes were previously incubated in serum-free and glucose-free DMEM medium and then treated with control vehicle PBS or Ric4 (100 µM) for 30 min. Proteins from subcellular fractions: N/ER (**C**), microsomal (**D**), plasma membrane (**E**), were isolated and the expression Glut4 was analyzed by Western blot. Images were quantified using ImageJ 1.49 software. The statistical comparisons were performed using analysis of variance (ANOVA), followed by ad-hoc Tukey’s test using GraphPad Prism software * *p* < 0.05; # *p* < 0.05; **** *p* < 0.0001.

**Figure 5 pharmaceutics-13-02175-f005:**
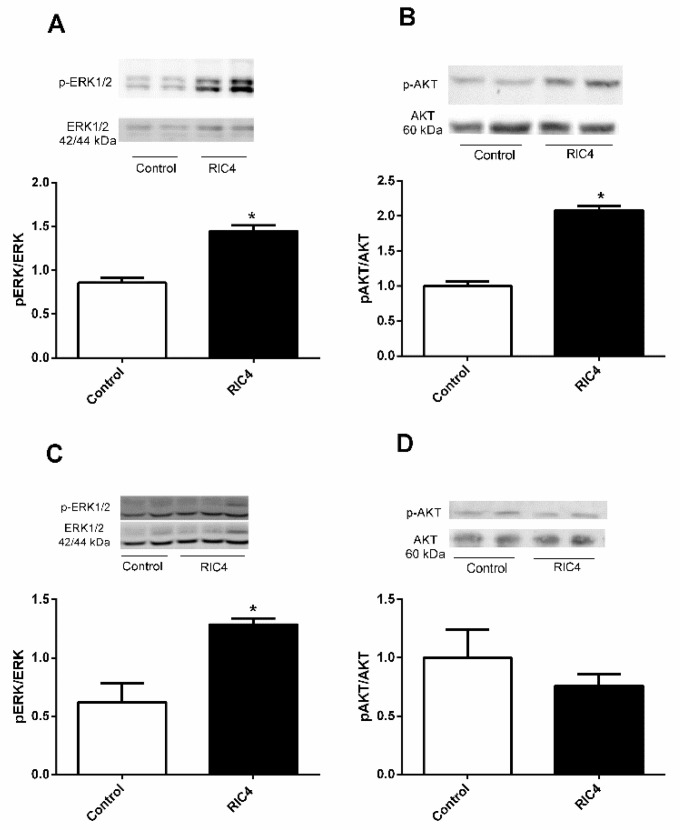
Western blotting for ERK and AKT in epididymal adipose and tibial muscle tissues following ip administration of Ric4 to WT mice. Mice were treated ip with vehicle (Control) or Ric4 (600 µg/kg). Tissues were collected 30 min after the Ric4 administration and processed for Western blots as detailed in Section 2.5. The phosphorylation of either ERK or AKT in epididymal adipose tissue (**A**,**B**) and tibial muscle tissue (**C**,**D**) were analyzed using antibodies anti-pErk (**A**,**C**) or anti-pAkt (**B**,**D**). Anti-total ERK, anti-total AKT, and β-actin were used for normalizing protein concentration. Imaging and band intensity measurements were performed using ImageJ 1.49 software. Data are representative of three independent experiments that produced similar results. The statistical comparisons were performed using Student’s *t*-test using GraphPad Prism software * *p* < 0.05.

**Figure 6 pharmaceutics-13-02175-f006:**
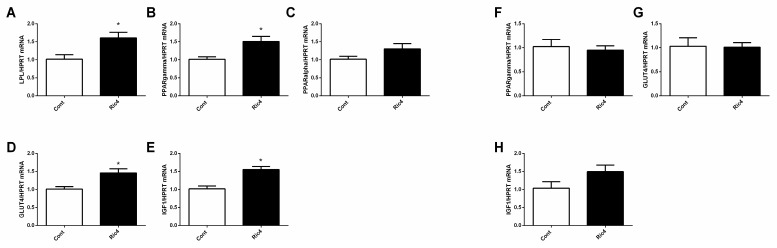
Ric4 stimulated mRNA expression levels of genes related to energy metabolism. WT mice were treated with vehicle (Control) or Ric4 (600 ug/kg, ip) and gene expression was evaluated by qPCR. LPL (**A**), PPARγ (**B**), PPARa (**C**), GLUT4 (**D**), and IGF1 (**E**), in epidydimal adipose tissue. PPARγ (**F**), GLUT4 (**G**), and IGF1 (**H**) in tibial muscle. Data are presented as mean + SEM. Statistical analyses were performed using Student’s unpaired *t*-test using GraphPad Prism software * *p* < 0.05.

**Figure 7 pharmaceutics-13-02175-f007:**
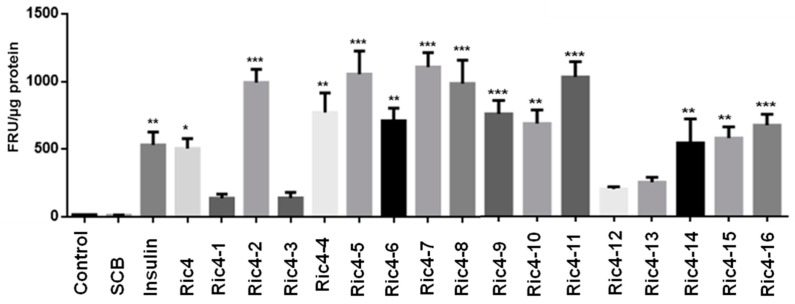
Glucose uptake in C2C12 cells. C2C12 cells were previously incubated in serum-free and glucose-free DMEM medium, and were treated with either insulin, Ric4, or indicated Ric4 derived peptide (100 µM; Table 3), in the presence of 2-[N-(7-Nitrobenz-2-oxa-1,3-diazol-4-yl)amino]-2-deoxy-d-glucose (2-NBDG; 80 µM). Insulin was used as the positive control. Before the assays, the culture medium was removed from each well and replaced with 100 μL of culture medium containing fluorescent d-glucose analog 2-NBDG (80 µM; for standardization concentrations of 25, 50, 80, and 100 µM were used, data not shown) in the absence or presence of the indicated compound (insulin, Ric4 or Ric4-1/16 derived peptides). Plates were incubated at 37 °C with 5% CO2 for a period of 30 min, and after that were lysed with 50 μL of 0.1 N NaOH and fluorescence of aliquots from the lysate was measured. The statistical comparisons were performed using Student’s *t*-test or analysis of variance (ANOVA), followed by ad-hoc Tukey’s test using GraphPad Prism software * *p* < 0.05; ** *p* < 0.001; *** *p* <0.0001.

**Figure 8 pharmaceutics-13-02175-f008:**
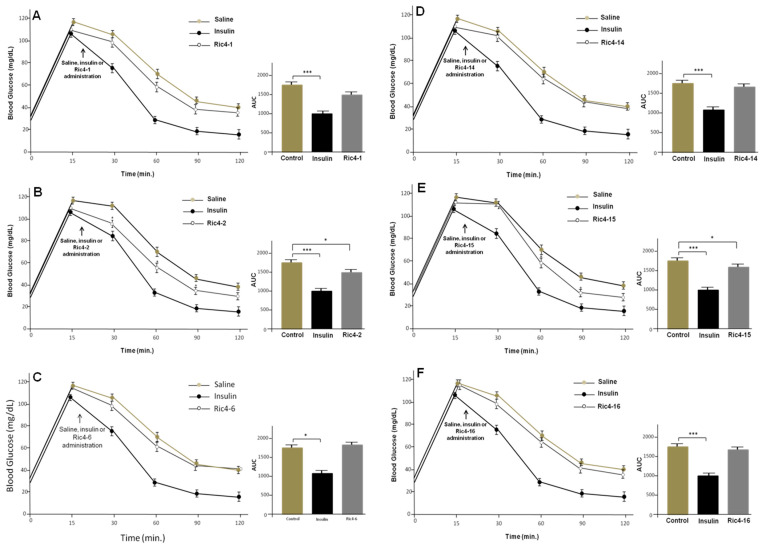
Glucose tolerance tests (GTT) in the mice administrated orally with Ric4-derived peptides. (**A**–**F**) Animals were fasted for 12 h before the experiments. Glucose (2 g/kg) was injected at time zero, and at time 20 min insulin (0.75 IU/kg, ip) was administrated. Indicated Ric4-derived peptides (600 µg/kg) were orally administrated by gavage. Control animals were administrated with saline. The statistical comparisons were performed using Student’s *t*-test or analysis of variance (ANOVA), followed by ad-hoc Tukey’s test using GraphPad Prism software * *p* < 0.05; *** *p* < 0.0001.

**Table 1 pharmaceutics-13-02175-t001:** Ric4 derived peptides.

Peptide	AA Sequence	pI (pH)	Net Charge (pH 7.0)
Ric4-1	Ac-LASVSTVLTSKYR	9.81	1.9
Ric4-2	Ac-LASVSTVLTSKYR-NH2	9.81	2
Ric4-3	ASVSTVLTSKYR	10.5	2
Ric4-4	SVSTVLTSKYR	10.41	2
Ric4-5	VSTVLTSKYR	10.46	2
Ric4-6	STVLTSKYR	10.41	2
Ric4-7	TVLTSKYR	10.4	2
Ric4-8	Ac-LASVSTVLTSKY	8.81	0.9
Ric4-9	Ac-LASVSTVLTSKY-NH2	8.82	1
Ric4-10	LASVSTVLTSKY	9.74	1
Ric4-11	LASVSTVLTSK	10.12	1
Ric4-12	LASVSTVLTS	3.72	0
Ric4-13	LASVSTVLT	3.69	0
Ric4-14	LASVSTVL	3.63	0
Ric4-15	Ac-LASVSTV[DLeu]TSKYR	9.81	1.9
Ric4-16	Ac-LASVSTV[DLeu]TSKYR-NH2	9.81	2

**Table 2 pharmaceutics-13-02175-t002:** InPeps Ric1-Ric4 effects on the expression of genes related to gluconeogenesis and skeletal muscle contraction.

					InPeps		
Gene Name	Symbol	Control	Insulin	Ric 1	Ric 2	Ric 3	Ric 4
Peroxisome proliferator activator receptor gamma	PPARγ	1	6.98 ± 1.23 **	1.76 ± 1.03	2.03 ± 0.98 *	1.12 ± 0.72	1.21 ± 1.02
cAMP responsive element binding protein 1	Creb1	1	6.48 ± 1.08 **	1.51 ± 1.01	0.96 ± 0.75	1.25 ± 0.94	1.03 ± 0.85
Aldolase A, fructose-bisphosphate	ALDOA	1	7.65 ± 0.96 **	3.81 ± 1.51	0.98 ± 1.53	1.01 ± 0.16	4.22 ± 0.74 **
Cytochrome c oxidase subunit IV isoform 1	Cox4i1	1	6.71 ± 1.24 **	1.31 ± 0.87	1.42 ± 0.98	1.02 ± 1.21	1.41 ± 1.12
Phosphoglycerate mutase 2	PGAM2	1	8.26 ± 0.99 **	4.05 ± 1.65*	1.03 ± 1.78	0.95 ± 0.57	2.65 ± 0.87 *
Troponin I, skeletal, fast 2	TNNI2	1	6.36 ± 1.13 **	1.97 ± 1.16	0.98 ± 0.89	1.12 ± 1.01	3.14 ± 0.65 **
Troponin I, skeletal, fast 3	TNNT3	1	6.22 ± 0.98 **	1.21 ± 1.17	3.08 ± 0.94 *	1.21 ± 1.11	2.38 ± 0.72 *
Small muscle protein X-linked	SMPX	1	5.96 ± 0.98 **	3.31 ± 1.67 *	1.05 ± 1.52	0.99 ± 0.32	3.21 ± 0.99 *
Myosin, light chain 1	MYL1	1	6.18 ± 1.07 **	3.76 ± 1.24 *	1.15 ± 0.87	1.06 ± 0.39	3.58 ± 1.07 **

Footnote: * *p* < 0.05, ** *p* < 0.01; compared to control.

**Table 3 pharmaceutics-13-02175-t003:** Evaluation of Ric4 and derived peptides hydrolytic stability.

Peptide.	Sequence	Liver Extract (30 µg)	Kidney Extract (3 µg)	THOP1	Nln
Bradykinin	RPPGFSPFR	25.4	17.4	11.93	14.51
RIC4	LASVSTVLTSKYR	62.4	22.6	2.22	4.69
RIC4-16	Ac-LASVSTV[DLeu]TSKYR-NH2	26.0	12.5	40.33	18.26
RIC4-2	Ac-LASVSTVLTSKYR-NH2	70.2	60.0	4.55	1.18

Footnote: All peptides were evaluated at initial concentration of 50 µM. Results shown % of peptide degraded during considering their 100% initial concentration. Results are the average from three independent determinations that varied less than 5% among each other. THOP1 (0.1 ng) or Nln (0.1 ng) were homogeneous recombinant enzymes and were prepared as previously described [37].

## Data Availability

WE hereby state that all data, tables and figures presented in the current manuscript were original, and were never submitted or published in another scientific journal.

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
