# Peer review of "New Intracellular Peptide Derived from Hemoglobin Alpha Chain Induces Glucose Uptake and Reduces Blood Glycemia"

_pharmaceutics, 2021, doi:10.3390/pharmaceutics13122175_

Round 1

Reviewer 1 Report

Reviewer comments:

Title: A new intracellular peptide derived from hemoglobin alphachain that induces glucose uptake and reduces blood glycemia

ID: pharmaceutics-1491448

(a) ARTICLE RANKING

* Fair.

(b) RECOMMENDATION

* Minor revision.

The article entitled, A new intracellular peptide derived from hemoglobin alphachain that induces glucose uptake and reduces blood glycemia” by Renée N. O. Silva 1, Ricardo P. Llanos 1, Rosangela A. S. Eichler 1, Thiago B. Oliveira, Fábio C. Gozzo, William T. Festuccia, and Emer S. Ferro. The article is interesting and important for developing new drugs and clinical therapeutic agents. However, clarity and English have to be considered for better quality. Need to improve connectivity and flow. Reduce complexity and make it interesting to readers. These suggestions are for all sections of the manuscript.

(c) Comments of Reviewer

TITLE:
*In title no need to use “A”, “that”

ABSTRACT:

* Intracellular peptides (InPeps) derived from proteasomal degradation of proteins within cells, “name the cell”. The first sentence of the abstract, needs clarification. (InPeps) the short form is unusual.

INTRODUCTION
*There is a need to address previous developments related to the field. Commercialization prospects of InPeps.

EXPERIMENTAL PROCEDURES

*(InPeps) synthesis protocol is missing.

*Why Cell viability experiments are performed, is their toxicity concerns.

* Western blotting, what are the reason to perform this experiment.

*Intracellular peptide stability is lengthy, if possible reduce to the length, express precisely.

RESULTS

*Scientific descriptions are necessary, most of the results are inadequately described, increase to the length.

*consider economic feasibility, large-scale production possibility.

DISCUSSION

*Address scientific developments so far made in the field, use maximum possible citations.

CONCLUSIONS
* Separate this section, Revise this section, and future implications.

Author Response

Dear Reviewer, we deeply acknowledge your time and efforts providing critical comments to improve the present manuscript. Please, find bellow our point-to-point answers addressing all your questions. 

 TITLE:
*In title no need to use “A”, “that”.

Answer: suggested modifications were made.

ABSTRACT:

  • Intracellular peptides (InPeps) derived from proteasomal degradation of proteins within cells, “name the cell”. The first sentence of the abstract, needs clarification. (InPeps) the short form is unusual.
  • Answer: suggested modifications were made accordingly. 

INTRODUCTION
*There is a need to address previous developments related to the field. Commercialization prospects of InPeps.

  • Answer: suggested modifications were made accordingly, please find such editions in the labeled version of the revised manuscript. 

EXPERIMENTAL PROCEDURES

*(InPeps) synthesis protocol is missing. Answer: Please find such editions in the labeled version of the revised manuscript.

*Why Cell viability experiments are performed, is their toxicity concerns. 

Answer: suggested modifications were made accordingly, please find such editions in the labeled version of the revised manuscript.

  • Western blotting, what are the reason to perform this experiment.
  • Answer: Western blots were performed to evaluate expression levels of specific proteins. We have provided a short explanation about these experiments in the text. Please find such editions in the labeled version of the revised manuscript.

*Intracellular peptide stability is lengthy, if possible reduce to the length, express precisely. 

Answer: suggested modifications were made accordingly, please find such editions in the labeled version of the revised manuscript.

RESULTS

*Scientific descriptions are necessary, most of the results are inadequately described, increase to the length.

Answer: We have improved the results description as suggested. However, because the manuscript  was prepared with both Results and Discussion sessions, most of the results discussion were presented at the Discussion session. Please, find such editions in the labeled version of the revised manuscript.

*consider economic feasibility, large-scale production possibility. 

Answer: We have provided a new paragraph at the Conclusion session, discussing about these issue. Please, find such editions in the labeled version of the revised manuscript.  

DISCUSSION

*Address scientific developments so far made in the field, use maximum possible citations. 

Answer: We have improved the discussion session appropriately. However, recently an extensive review article was published presenting in details all InPeps with potential pharmaceutical application (De Araujo et al, Biomolecules, 2019). We have now mentioned this review on the discussion session.   

CONCLUSIONS
* Separate this section, Revise this section, and future implications.

Answer: suggested modifications were made accordingly, please find such editions in the labeled version of the revised manuscript.

Reviewer 2 Report

Figure 1 legend is incomplete. These is no detail about the upper panned of images 0d, 5d, and 7d.

In result section, authors have mentioned “Peptides Ric1, Ric 2 and Ric4, but not Ric3, increased the expression of different genes related to glucose metabolism via glycolysis (aldolase and phosphoglycerate mutase) and skeletal muscle contraction (Table 1)”. However, as compared to Ric3, ALDOA gene expression is less for Ric2 which is contradicting the above-mentioned statement.

In addition, in table 1, there is no clarification on what control used and why its value is same (1) in all cases. Kindly specify which gene corresponds to skeletal muscle contraction.

Rephrase the below mentioned sentence.

“Insulin, Ric2 and Ric4 similarly increased phosphorylation of Erk”

Experimental section-Intracellular peptide stability

Replace “or” with “for” in the below mentioned sentence

“each peptide was individually incubated or 20 min”

Why insulin was used at lower concentration (100nM) as compared to the test peptides (100µM) for activation of Erk or Akt experiments.

In figure 2B, the GAPDH bands appears like a trail.

In results section, below mentioned paragraph is mentioned twice.

In addition to gene expression, we next investigated the effects of Ric1, Ric2, Ric3 and Ric4 on signaling through extracellular signal-regulated kinase 1/2 (Erk1/2, Erk phosphorylation Thr202-Tyr204and/or Akt (phosphorylation at Ser473), in differentiated C2C12 cells (Fig. 2). Insulin, Ric 2 and Ric 4 similarly increased phosphorylation of Erk (Fig. 2, A). Insulin, Ric2, Ric3 and Ric4 also increased phosphorylation of Akt (Fig. 2, B). Ric1 neither stimulate phosphorylation of Erk nor Akt (Fig. 2). Erk and Akt signaling pathways can be activated by insulin[37], whereas activation of Akt signaling pathway have been most frequently associated with Glut4-induced glucose uptake[38].

The below sentence is not giving a clear information about whether the glucose level reaches to 400 mg/dL after glucose injection of test specimen injection. Rephrase it.

“After 20 min of glucose injection, WT mice blood glucose levels reached 400 mg/dL, when these animals received an ip administration of either saline, insulin (0.75 IU/Kg), Ric2 (600 µg/Kg) or Ric4 (600 µg/Kg)”   

Below mentioned sentence is not clear

“Next, in differentiated C2C12 cells similar to insulin Ric4 was observed to stimulate glucose uptake both in differentiated C2C12 (Fig. 4 A; left panel, using 3H-glucose; right panel, using 2-NBDG H-glucose) and adipose tissue explants obtained from C57BL6N WT mice (Fig 4, B)”

Replace it with “Similar to insulin, Ric4 was observed to stimulate glucose uptake both in differentiated C2C12 (Fig. 4 A; left panel, using 3H-glucose; right panel, using 2-NBDG H-glucose) and adipose tissue explants obtained from C57BL6N WT mice (Fig 4, B)”

Replace “Similar to insulin Ric4 induced translocation of Glut4 to plasma membrane (Fig. 4, E)”

With “Similar to insulin, Ric4 induced translocation of Glut4 to plasma membrane (Fig. 4, E)”

A total of 16 derivates of the Ric4 peptides (Ric4-1 to Ric4-16) were designed and examined for glucose uptake. Structural modifications like N- and C-terminal modification and L to D-amino acid replacement are the well-known approaches to provide extra stability against enzymatic degradation. There is no explanation about other rational of the synthesis of other derivatives.

Identification of the enzymatic susceptibility of the amide bond between Leu8-Thr9 is very interesting finding. Considering the importance of the data in the context of the manuscript, the stability and fragmentation pattern data acquired by LC-MS/MS should be included in the supporting information.

Author Response

Dear reviewer, we deeply acknowledge your critical contribution to improve our present manuscript. Please, find bellow point-to-point answers to your comments. 

Comments and Suggestions for Authors

Figure 1 legend is incomplete. These is no detail about the upper panned of images 0d, 5d, and 7d.

Answer: modifications were made to improve the legend of  Figure 1. Please, find such edition on the labeled version of the revised manuscript. 

In result section, authors have mentioned “Peptides Ric1, Ric 2 and Ric4, but not Ric3, increased the expression of different genes related to glucose metabolism via glycolysis (aldolase and phosphoglycerate mutase) and skeletal muscle contraction (Table 1)”. However, as compared to Ric3, ALDOA gene expression is less for Ric2 which is contradicting the above-mentioned statement. In addition, in table 1, there is no clarification on what control used and why its value is same (1) in all cases. Kindly specify which gene corresponds to skeletal muscle contraction.

Answer: Increased gene expression refers to control 1, which is relative the level of expression of GAPDH gene (100% = control 1). Modifications were made on the text to explain these results, including the genes related to muscle contraction. Please, find such edition on the labeled version of the revised manuscript.

Rephrase the below mentioned sentence.

“Insulin, Ric2 and Ric4 similarly increased phosphorylation of Erk”

Answer: Please, find such edition on the labeled version of the revised manuscript.

Experimental section-Intracellular peptide stability

Replace “or” with “for” in the below mentioned sentence. “each peptide was individually incubated or 20 min” 

Answer: Please, find such edition on the labeled version of the revised manuscript.

Why insulin was used at lower concentration (100nM) as compared to the test peptides (100µM) for activation of Erk or Akt experiments.

Answer: Insulin is more potent than Ric4 to induce glucose uptake, also that is the standard dose of insulin. 

In figure 2B, the GAPDH bands appears like a trail. 

Answer: Thank you for pointing this issue. We have removed the GAPDH from the figure. It do not compromise the results, because the level of phosphorylation of Erk or Akt induced by Ric4 or insulin were made calculated based on the pErk/total Erk or pAkt/total Akt. Please, find such edition on Fig. 2 of the revised version of the manuscript.

In results section, below mentioned paragraph is mentioned twice.

In addition to gene expression, we next investigated the effects of Ric1, Ric2, Ric3 and Ric4 on signaling through extracellular signal-regulated kinase 1/2 (Erk1/2, Erk phosphorylation Thr202-Tyr204and/or Akt (phosphorylation at Ser473), in differentiated C2C12 cells (Fig. 2). Insulin, Ric 2 and Ric 4 similarly increased phosphorylation of Erk (Fig. 2, A). Insulin, Ric2, Ric3 and Ric4 also increased phosphorylation of Akt (Fig. 2, B). Ric1 neither stimulate phosphorylation of Erk nor Akt (Fig. 2). Erk and Akt signaling pathways can be activated by insulin[37], whereas activation of Akt signaling pathway have been most frequently associated with Glut4-induced glucose uptake[38].

Answer: Thank you for pointing this problem, we have edited the  paragraph accordingly. Please, find such edition on the labeled version of the revised manuscript.

The below sentence is not giving a clear information about whether the glucose level reaches to 400 mg/dL after glucose injection of test specimen injection. Rephrase it.

“After 20 min of glucose injection, WT mice blood glucose levels reached 400 mg/dL, when these animals received an ip administration of either saline, insulin (0.75 IU/Kg), Ric2 (600 µg/Kg) or Ric4 (600 µg/Kg)”   

Answer: Thank you for pointing this problem, we have edited the  sentence accordingly. Please, find such edition on the labeled version of the revised manuscript.

Below mentioned sentence is not clear

“Next, in differentiated C2C12 cells similar to insulin Ric4 was observed to stimulate glucose uptake both in differentiated C2C12 (Fig. 4 A; left panel, using 3H-glucose; right panel, using 2-NBDG H-glucose) and adipose tissue explants obtained from C57BL6N WT mice (Fig 4, B)”

Replace it with “Similar to insulin, Ric4 was observed to stimulate glucose uptake both in differentiated C2C12 (Fig. 4 A; left panel, using 3H-glucose; right panel, using 2-NBDG H-glucose) and adipose tissue explants obtained from C57BL6N WT mice (Fig 4, B)”

Replace “Similar to insulin Ric4 induced translocation of Glut4 to plasma membrane (Fig. 4, E)”

With “Similar to insulin, Ric4 induced translocation of Glut4 to plasma membrane (Fig. 4, E)”

Answer: Thank you for helping, we have edited the  sentences accordingly. Please, find such edition on the labeled version of the revised manuscript.

A total of 16 derivates of the Ric4 peptides (Ric4-1 to Ric4-16) were designed and examined for glucose uptake. Structural modifications like N- and C-terminal modification and L to D-amino acid replacement are the well-known approaches to provide extra stability against enzymatic degradation. There is no explanation about other rational of the synthesis of other derivatives.

Answer: We have provided an explanation for the rational design of additional Ric4 derivatives. "The rational to design additional peptides was to identify a minimal Ric4-derived sequence retaining the ability to induce glucose uptake. Thus, Ric4 original sequence was successively shortened from amino and/or carboxyl terminal" Please, find such edition on the labeled version of the revised manuscript.

Identification of the enzymatic susceptibility of the amide bond between Leu8-Thris very interesting finding. Considering the importance of the data in the context of the manuscript, the stability and fragmentation pattern data acquired by LC-MS/MS should be included in the supporting information.

Answer: We thank the reviewer for these comments. Unfortunately we were unable to find the backup files relative to the MS/MS data of these sequencing. Some time ago one of our external drives storing data was damaged after dropping on the floor. Most possibly the present MS/MS data was stored inside of that drive. Because of that, we have modified the text accordingly to avoid mentioning that LC/MS/MS experiments were performed.  However, we have provided a supplemental figure (Supplemental Fig. 1) with the HPLC data that illustrates these experiments, showing the greater enzymatic stability of the DLeu peptide that was synthesized. We deeply apologize about the lack of these data.   
